# Machine Learning to Predict the Adsorption Capacity of Microplastics

**DOI:** 10.3390/nano13061061

**Published:** 2023-03-15

**Authors:** Gonzalo Astray, Anton Soria-Lopez, Enrique Barreiro, Juan Carlos Mejuto, Antonio Cid-Samamed

**Affiliations:** 1Universidade de Vigo, Departamento de Química Física, Facultade de Ciencias, 32004 Ourense, Spain; 2Universidade de Vigo, Departamento de Informática, Escola Superior de Enxeñaría Informática, 32004 Ourense, Spain

**Keywords:** microplastics, adsorption capacity, machine learning, random forest, support vector machine, artificial neural network, prediction

## Abstract

Nowadays, there is an extensive production and use of plastic materials for different industrial activities. These plastics, either from their primary production sources or through their own degradation processes, can contaminate ecosystems with micro- and nanoplastics. Once in the aquatic environment, these microplastics can be the basis for the adsorption of chemical pollutants, favoring that these chemical pollutants disperse more quickly in the environment and can affect living beings. Due to the lack of information on adsorption, three machine learning models (random forest, support vector machine, and artificial neural network) were developed to predict different microplastic/water partition coefficients (log *K*_d_) using two different approximations (based on the number of input variables). The best-selected machine learning models present, in general, correlation coefficients above 0.92 in the query phase, which indicates that these types of models could be used for the rapid estimation of the absorption of organic contaminants on microplastics.

## 1. Introduction

Since the appearance of plastics, their production has grown exponentially in recent decades, and due to their versatility, they are used in different fields, such as packaging, building, or electronic industries, among others [1]. The use of microbeads or nanobeads based on plastic polymers (e.g., Bisphenol-A diglycidyl ether, polyetheramine, and Polyvinyl alcohol) to functionalize and decorate CNTs has been reported to improve their shielding against electromagnetic interference, and the electrical and mechanical properties of elastic, functional composites can also be improved by interlacing the beaded and coated fibers into a smart tissue [2,3]. The transformation of plastics into microplastics (MPs) and then into nanoplastics (NPs) through fragmentation makes the presence of micro- and nanoplastics (MNPs) in both water sources and our planet’s agroecosystems a worldwide concern [4,5,6,7]. In this sense, and as reported by Matthews et al. (2021), microplastics are plastic fragments of less than 5 mm, and the most commonly accepted size for nanoplastics is that falling within the range of 1–1000 nm [8].

The cycle from the production of plastics to their entry into the environment includes different stages, as reported by Woods et al. (2021) [9]: production for textile manufacturing and use, tires use, or packaging production, among others [10]. Additionally, there are different pollution sources by MPs; due to this, they can be differentiated between primary and secondary sources [11]. Regarding the primary MPs, they are commonly generated during the manufacturing of different products or the fabrication of microbeads or microfibers; while the secondary ones are formed from larger plastic litters due to degradation processes by external factors (chemical, physical or biological) [11].

Once released into the aquatic environment, it has been reported that microplastics can be vectors for persistent organic pollutants (POPs) [12,13]. Chemical contaminants can be captured on the surfaces of microplastics and nanoplastics due to, for example, their surface charge [14]. 

Currently, micro- and nanoplastics can be found in terrestrial and aquatic environments, are able to affect a large number of organisms [8], and can be considered Trojan horses, as reported by Katsumiti et al. (2021) and Hu et al. (2022) [15,16].

The adsorption capacity of organic polluting substances among microplastics and water could be understood as an equilibrium partitioning coefficient (*K*_d_) [17]. Since, according to the authors [17], the absorption data are nowadays limited, it would be interesting to dispose a method to predict the *K*_d_ values under various conditions. For this reason, the application of the quantitative structure–property relationship (QSPR) together with machine learning (ML) techniques would be a possible alternative for the determination of these values. 

Machine learning is one of the subsets that includes artificial intelligence (AI) which consists of training machines that are capable of imitating the intellectual ability of the human being based on the knowledge and experience learned [18]. Most existing machine-learning methods involve the use of a set of training cases, where each case contains input and output labels [19]. Then, the method tries to predict the output labels using the inputs from the training data [19]. Due to these characteristics, this type of supervised methods are suitable for the predictive modeling required in this research. The three supervised learning models used in this research will be briefly presented below. They were chosen because they have the common objective of building models that can generalize patterns from input data.

### 1.1. Random Forest

The first of the selected ML models is a random forest (RF) model, which is composed of decision trees that can be applied for regression and classification purposes [20]. These decision trees can be considered as among the main methods for solving real problems [21]. Random forests are an ensemble machine learning method that has been proposed by Breiman (2001) [22] that can overcome the instability and overfitting problems that arise when only using a single decision tree [23]. When working on regression and classification mode, the random forest uses the bootstrap samples of the original training data to develop and train each decision tree, generating a series of decision trees [24]. Therefore, the random forest involves the development of different decision trees using random subsets of the original training data [23]. Each decision tree in RF starts at a root node and recursively splits into two subnodes based on minimizing the mean square error [25]. The predictions of each individual tree are aggregated to form the final prediction. For regression tasks with quantitative data, the predictions are averaged, whereas for classification tasks with qualitative data, a voting process is carried out [20,21,26]. RF is generally considered a robust method that can achieve good results compared to different regression algorithms [27].

Random forest models can be used in different research fields, such as: cloud computing to develop a DDoS-attack-detection method [20], chemistry to determine molecular electronic transitions [25], or in farming to predict regional and local-scale wheat yield [28], among others.

### 1.2. Support Vector Machine

The second of the ML models developed in this research is the support vector machine (SVM). According to Geppert et al. (2010) [29], this method became popular during the 1990s based on the work carried out by Cortes and Vapnik (1995) [30] and, as reported by Rodríguez-Pérez et al. (2017) [31], has become more and more popular. A support vector machine is based on statistical learning theory [32] and can be used for regression and classification tasks [32,33]. According to Houssein et al. [33], the support vector machine can work with linear and nonlinear problems. When working in classification mode, the main aim of an SVM is to identify a hyperplane within an N-dimensional space to clearly classify the data points [34]. As Sareminia (2022) reports, different hyperplanes could be chosen for two different data classes, but the aim of the SVM is to locate the plane with the largest margin, which is the maximum gap between both classes’ data points [34]. Whether the problem is linear or nonlinear, the support vector machine separates the data into two classes by mapping the information into spaces with dimensions greater than two [35]. On the other hand, Rodríguez-Pérez et al. (2017) [31] report that SVM can also be used in the regression model (support vector regression, SVR) to predict numerical property values [36,37]. In this type of SVM model, a different function is derived from the training data for predicting numerical values [31].

Support vector machines can be applied in different fields such as chemistry to identify the polar liquids [38], energy storage to self-discharge prediction in batteries of lithium-ion [39], or in medicine to diagnose breast cancer [40], inter alia.

### 1.3. Artificial Neural Networks

The last ML models carried out in this research were models based on artificial neural networks (ANNs). An artificial neural network is a well-documented/known artificial intelligence model [41] that can be defined as a mathematical model which is inspired by the behavior of biological neurons [21,41]. As reported by Paturi et al. (2022) [42], McCulloch and Pitts (1943) [43] were the first who could explain the logical relationship that exists between the neural events of the nervous system. This imitation of the behavior of biological neurons can be learned through a process of backpropagation [44]. 

ANN is a powerful tool to find relationships between data, in this case, input and output data [41], and can be used to solve complex problems in optimization, clustering, or prediction, among others [45]. ANN is formed by units (neurons) that are organized into different layers [44]. A neuron performs two functions: collecting the inputs and producing an output [45]. An ANN architecture is usually made up of three elements, a first layer (named the input layer), a second layer (known as the hidden layer), and a final layer (named the output layer) [46]. One of the existing neural network types, the multi-layer perceptron network (MLP), possesses one or more hidden layers [42,46], and, in principle, and according to Saikia et al. (2020) [47], it is possible to approximate any continuous function with only one hidden layer [48]. According to [47], an artificial neural network is a popular ML procedure due to its capacity for complex nonlinear function modeling. The neurons number located in the intermediate layer can be established by trial and error [46,49].

To find the relationship between the input and the output data, the artificial neural network must necessarily be subjected to a training process using the database containing both input and output data [41]. Following Niazkar and Niazkar (2020) [41], the first layer contains neurons associated with the input vector; a hidden layer connects the input neurons and the output neuron/neurons and turns the input data into the correspondent output data. Finally, the output layer presents the neuron/neurons associated with the output vector. Each processing neuron is typically modeled as a computational unit that takes the input value and multiplies it by the learned importance of the connection, also called weight, and the result and bias are processed by an activation function and provide an output in the neuron [45]. In fact, there are different activation functions, such as sigmoid or Gaussian, among others [46]. 

Finally, artificial neural networks can be used in engineering to predict the building construction time and cost [50], om water management to model and predict the amount of salt removed by the capacitive deionization method [51], or in biotechnology to optimize the parameters in *Ganoderma lucidum* residue aerobic composting process [52].

Therefore, this research aimed to apply machine learning models (RF, SVM, and ANN) to predict the adsorption capacity of MPs ((polyethylene (PE), polystyrene (PS), polypropylene (PP)) in different waters using different configurations of input variables (molecular mass (*M*’_w_), n-octanol/water distribution coefficient under special pH condition (log D), and other quantum chemical descriptors) obtained from the literature [17]. These computational models will enable the quick prediction of the adsorption capacity of organic pollutants onto these three types of microplastics in water environments.

## 2. Materials and Methods

### 2.1. Experimental Data Used

The data used for the development of the different ML models were extracted from the work developed by Li et al. (2020) [17]. Li et al. (2020) [17] also used different articles reported in the literature to obtain data. These articles can be consulted in Table 2 of the research paper of Li et al. (2020).

In their study and accompanying supplementary material, Li et al. (2020) provided: (i) the n-octanol/water distribution coefficient under special pH conditions (log D); (ii) the molecular mass (*M*′_w_); and (iii) six different quantum chemical descriptors that allow the modeling of the microplastic/water partition coefficients (log *K*_d_) for diverse organics between and polyethylene/seawater–freshwater–pure water, polystyrene/seawater, and polypropylene/seawater [17]. The quantum chemical descriptors calculated by Li et al. (2020) were: (i) molecular volume (*V*′); (ii) the most negative atomic charge (*q*^−^); (iii) the most positive atomic charge on the H atom (*q*H^+^); (iv) the ratio of average molecular polarizability and molecular volume (π) and the covalent; (v) basicity (ε_β_); and (vi) acidity (ε_α_).

In the present research work, two approximations were carried out. The first is using the same variables that the researchers used to develop their models (Type 1) [17]. On the other hand, due to the authors’ data having 8 different input variables, models that included the maximum number of these variables (Type 2) were developed to improve the previous models (Type 1). Table 1 shows the variables selected for each selected model.

The database was divided into three datasets. In this sense, the cases used by Li et al. (2020) to develop the models were used to generate two groups, a training group to elaborate different ML models and another group, the validation group, to select the best model (considering to the RMSE value in the validation phase). The query group (the cases reported by Li et al. (2020) as test cases in their Table 2) were used to check the adjustments provided by the different ML models. 

### 2.2. Models Implemented

#### 2.2.1. Random Forest Models

Random forest model have been successfully applicated in fields related to this research, for example, to identify and monitor different microplastics in environmental samples [53]. Hufnagl et al. (2019) developed a methodology to discriminate five different polymers (polyethylene, poly(methyl methacrylate), polypropylene, polystyrene, and polyacrylonitrile) and determine their abundance and size distribution [53]. Later, some previous authors extended the previous research to develop a model capable of differentiating more than 20 types of polymers [54].

The RF models (Figure 1A) were carried out using different parameter combinations. The following parameters were studied: the number of trees (1 to 100 using 99 steps in linear scale), maximum depth (1 to 100 using 99 steps in linear scale), and prepruning (false or true). All models were developed using the least square criterion.

#### 2.2.2. Support Vector Machine Models

Support vector machine models have also been successfully used in related fields. An example of this is the research carried out by Yan et al. (2022) [59]; the aim was to develop an ensemble machine learning method capable of classifying and identifying MPs by attenuated total reflection–Fourier transform infrared spectroscopy (ATR-FTIR) data. On the other hand, Bifano et al. (2022) [60] developed a method based on a support vector machine to detect polypropylene and polyolefin in water using electrical impedance spectroscopy.

In the research presented in this article, the SVM models (Figure 1B) were implemented using the LibSVM learner developed by [61,62]. The following parameters combinations were studied: the SVM type (ε-SVR or ν-SVR); γ was studied between ≈2^−20^ and 2^8^ using 28 steps in linear or logarithmic scale; and C between ≈ 2^−10^ and 2^20^ using 30 steps in linear or logarithmic scale (SVM and SVM_log_). These values are an extension of the proposed values of Hsu et al. (2016) [63]. In addition to using the database in their real-scale, they were also normalized in the interval [−1,1] (first just normalizing the input variables (SVM_n_ and SVM_n log_) and then normalizing the input and the output variables (SVM_n2_ and SVM_n2 log_). The normalization was applied to the training input data, and later applied to the other phases. After the model selection, the output data were de-normalized to allow the real-scale comparison between all developed models

#### 2.2.3. Artificial Neural Network Models

Artificial neural network models have been used to categorize microplastic contamination in the soil using infrared spectroscopy [64]. On the other hand, ANN has also been used successfully to determine the sorption capacity of heavy metal ions onto microplastics [65]. In this sense, Guo and Wang (2021) developed an ANN model using data from the literature and were able to determine the sorption capacity of different heavy metal ions onto microplastics in global environments with correlation coefficients greater than 0.92. 

In the research presented in this article, the ANNs (Figure 1C) were developed with one single hidden layer. The hidden neurons were analyzed in the range between 1 and 2n+1, where n is the input neurons number. The training cycles were studied between 1 and 131072 using 17 steps in linear or logarithmic scale (ANN_lin_ and ANN_log_). In addition, the decay was studied in mode true or false. The neural net operator to develop the ANN models scaled the values between −1 and 1 [66].

#### 2.2.4. Statistics Used to Analyze the Models

Different statistical parameters were used to evaluate the ML models implemented in this research. In this sense, the correlation coefficient (r), the root mean square error (RMSE), and the mean absolute percentage error (MAPE, expressed in %) were calculated (for training, validation, and query phases).

The best model for each ML approach was chosen considering the root mean square error for the validation phase. Once each best ML model was chosen, they were compared using the query data.

#### 2.2.5. Equipment and Software Used for the Development of the Models

The developed ML models were implemented in two computers; the first, an Intel^®^ Core™ i9-10900 at 2.80 GHz with 64GB RAM and Windows 10 Pro 21H1, and the second, an AMD Ryzen 7 3700X 8-Core at 3.60 GHz with 32 GB RAM and Windows 11 Pro 21H2.

The data used in this research were collected from Li et al. (2020) [17] using Microsoft Excel 2016 from Microsoft Office Professional Plus 2016. The ML models (RF, SVM, and ANN) were developed using two versions of RapidMiner Studio 9.10.001 and 9.10.011 software (Educational and a free). Figures were drawn with Microsoft PowerPoint 2016 from Microsoft Office Professional Plus 2016 and SigmaPlot v. 13.0 from Systat Software, Inc. (Palo Alto, Santa Clara, CA, USA).

## 3. Results and Discussion

The following sections analyzed the results obtained by the different machine learning methods for each of the analyzed assumptions.

### 3.1. ML Models Using Input Variables Type 1

Table 2 shows the adjustments obtained for the selected machine learning models to predict log *K*_d_, developed with the same variable combination used by Li et al. (2020) [17].

The first models (PE/seawater) correspond with ML models to predict the adsorption capacity for polyethylene in seawater. In this case, the three best-selected models (each according to their RMSE value for the validation phase) can be seen. The model with the best adjustments is the artificial neural network (ANN_log_) model (0.236), followed by the support vector machine (SVM_n2 log_) model (0.248), and finally, the random forest model (0.380). As can be seen, the three models present very high correlation coefficients for the validation phase, equal to or greater than 0.988; in addition, the mean absolute percentage error remains low, between 4.42% and 7.48%.

The good adjustments shown in the validation phase can also be observed in the training phase, where the values of RMSE remain similar to those of the validation phase, except for the random forest model, where the RMSE and MAPE values grow to 0.525 and 18.67%, respectively. As can be seen in the case of the query phase, the model that provided the best result in the validation and training phases, the ANN model, presents the worst results in terms of RMSE and MAPE (0.561 and 23.33%, respectively), despite it maintaining a high coefficient of correlation (0.979). The other two models, the support vector machine and the random forest model present slightly higher errors in terms of RMSE than those presented in the validation phase (0.357 and 0.523, respectively). 

Given these results (Table 2), it can be said that the three models generally show a good performance, although, for the query phase, the errors slightly increase. Despite this, the errors, in terms of RMSE, remain below the test error reported by Li et al. (2020) (0.752) for the model developed with these three input variables (log *D*, *ε*_α_, and *ε*_β_).

The second group of models (PE/freshwater) corresponds to machine learning models that predict the adsorption capacity for polyethylene in freshwater. In this case, it can be seen, in the case of the validation phase, that the errors made in terms of RMSE are closer to each other compared to the model’s behavior in the previous block. In this case, it can be seen that the worst model corresponds to the artificial neural network (ANN_lin_) model that presents an RMSE of 0.865, followed by the support vector machine (SVM_n log_) model with a value of 0.770, with the best model being the random forest, which has a root mean square error of 0.744. In this case, it can be seen that the mean absolute percentage errors exceed those obtained by the ML models of the first block, varying between 11.14% and 13.67%. 

Regarding the training phase, it can be seen that the validation phase adjustments are improved in a significant way, presenting RMSE values falling between 0.489 and 0.549. Regarding the query phase, it can be seen that the root mean square error remains at acceptable levels, corresponding to mean absolute percentage errors between 7.23% and 10.46%. The best model for the validation phase (RF with RMSE of 0.744) presents the worst results for the query phase (RMSE of 0.565) and vice versa; the best model of the query phase (ANN with RMSE of 0.464) is the worst model in the validation phase (RMSE of 0.865).

Despite these behaviors, the three selected models have suitable adjustments for all phases (Table 2). If these models are compared with the model developed by Li et al. (2020), it can be seen that all of them improve the adjustments in terms of the RMSE value in the test phase (0.661 vs. 0.464, 0.475, and 0.565) for the model developed with this input variable (log *D*).

The two following groups (PE/pure water—1; and PE/pure water—2) correspond to the machine learning models developed to determine the adsorption capacity for polyethylene in pure water. In this case, two blocks were developed because Li et al. (2020) presented two different approaches, one using two input variables (PE/pure water—1 with log *D* and *M*′_w_) and the other one using only one input variable (PE/pure water—2 with log *D*). 

In our research, for the model development with two input variables (PE/pure water—1), the case of 17α-ethinyl estradiol was not considered because the authors did not report the experimental log *K*_d_ value, so this model lacks this case. As expected, the models offer different results depending on the input variables. When two input variables are used, the model that presents the best results for the validation phase is the support vector machine (SVM_n log_) model, while when only one input variable is used, the best model is the random forest. It can be seen that the use of two input variables improves the adjustments in the training and validation phases (except for the RF model). For the query phase, the adjustments remain practically unchanged, except for the case of the ANN (ANN_lin_) model, where the error, in terms of RMSE, drops from 0.729 to 0.431. As can be seen, the models developed with two input variables present low mean absolute percentage errors between 2.06% and 3.92% for the validation phase. This behavior worsens slightly for the training phase, passing to 4.92% and 5.93% for the ANN and SVM models, respectively, and 11.28% for the RF model. On the other hand, in the case of the query phase, the MAPE values are between 6.90% and 12.21%. Despite the increase in both the RMSE and the MAPE values, these models developed with two variables seem to behave adequately to predict log *K*_d_.

The models developed to predict the adsorption capacity for polyethylene in pure water (PE/pure water—2) present, in general, slightly lower adjustments than those obtained by PE/pure water—1). In this case, the best model, considering the value of the root mean square error in the validation phase, is the random forest model, which presents an RMSE of 0.132. This model presents, in the query phase, an increase in its RMSE value (0.526). The other two models, namely the SVM (SVM_n2 log_) model and the ANN (ANN_lin_) model, present an RMSE value of 0.439 and 0.431 for this phase, slightly improving the results of the RF model for this phase.

According to these results (Table 2), it can be said that the SVM and ANN models for PE/pure water—2 show good performances in terms of RMSE and improve the adjustment of RMSE value for the test phase (0.471) provided by the model developed by Li et al. (2020) using only one input variable (log *D*).

Before continuing, it is necessary to emphasize that all the machine learning models developed to predict the adsorption capacity for PE in the different water samples present, in terms of mean absolute percentage error for the query phase, adequate values, generally, below 10%. In other cases, the value is slightly higher (SVM for PE/freshwater and ANN for PE/pure water—1), and in others, the difference is more significant, for example, for the models intended to predict log *K*_d_ in seawater, which present errors between 13.24% and 23.33%.

The following models (PP/seawater) correspond to the models developed to predict the adsorption capacity of polypropylene in seawater. Based on the results provided in the validation phase, it can be said that the best model corresponds to the random forest model (0.199), followed by the SVM (SVM_log_) model with an RMSE of 0.244 and, finally, the artificial neural network (ANN_lin_) model (0.270). The other statistics parameters of the validation phase show favorable behavior with MAPE values below 9% and with correlation coefficients above 0.980. For the training phase, the adjustments are similar to the validation phase, although an increase in the MAPE value of the random forest model is observed; even so, it remains below 10%.

For the query phase, an inconsistent behavior can be observed. Thus, in the case of the RF model and the ANN models, it can be observed that the statistics remain close to the values of the training and the validation phase, while the SVM model suffers an increase in terms of RMSE that makes this statistic parameter reach a value of 0.779, lowering its correlation coefficient to 0.817. 

Given these results (Table 2), it can be said that the RF and ANN models can perform prediction tasks correctly. These two models present lower RMSE values (0.298 and 0.307) than the model proposed by Li et al. (2020) in the test phase (0.369), which was developed with two input variables (log *D* and *ε*_β_). The SVM model presents high generalization errors, which imply that it should not be used for prediction tasks. It should be noted that this SVM model, which is the one with the lowest error for the validation phase among all the SVM models developed, is the one with the highest error for the query phase. Other SVM models with close RMSE values in the validation phase (0.255 and 0.262) subsequently showed a better result in the query phase (0.287 and 0.266, respectively). 

Finally, the last group of models (PS/seawater) developed corresponds to the machine learning models aiming to predict the adsorption capacity for polystyrene in seawater. Based on the results shown in Table 2, and taking into account the value of RMSE for the validation phase, it can be stated that the model presenting the best behavior in this phase is the support vector machine (SVM_n2 log_) model (0.524), followed by the artificial neural (ANN_lin_) network (0.643) and the random forest model (0.794). Based on the results presented by the mean absolute percentage error, it can be affirmed that these models destined to predict the adsorption for PS in seawater are the models that present the worst adjustments for the validation phase, varying between 14.61% and 21.69%. Despite this, the correlation coefficients remain high, with values greater than 0.960, except for the random forest model, whose correlation coefficient falls to 0.883. For the query phase, the values in terms of RMSE remain close, except for the random forest model, keeping the MAPE values above 15.1%.

Taking into account the results shown in Table 2, it can be concluded that the models predicting the adsorption capacity for PS in seawater do not present, in general, good results, except for the SVM model, which improves the RMSE value for the test phase (0.714) of the model developed by Li et al. (2020) with two input variables (log *D* and π).

Taking into account the fact that the results obtained by the machine learning models used the same variables as Li et al. (2020), it can be said that, in general, the ML models improve the results obtained by Li et al. (2020). However, these types of ML models often need a large number of experimental cases and input variables to correlate the desired variable. Therefore, in this research, in addition to developing ML models with the variables used by Li et al. (2020), other ML models have been developed with more input variables. This is possible because Li et al. (2020) reported eight different input variables; therefore, the results obtained by the models with the input variables selection Type 2 are shown below (Table 3).

### 3.2. ML Models Using Input Variables Type 2

Table 3 shows the adjustments obtained for the machine learning models developed with the input variables combination Type 2 using all the available input variables (except for the cases in which the variable *q*H^+^ is not possible).

The first models (PE/seawater) correspond with to ML models to for predicting the adsorption of polyethylene in seawater. Unlike the Type 1 models for PE/seawater where three input variables, log *D*, *ε*_α_ and *ε*_β_ were used, in this new PE/seawater model, seven input variables were used (log *D*, *M*′_w_, *ε*_α_, *ε*_β_, *q^−^*, *V*′, *π*). It can be observed (Table 3), based on the RMSE value for the validation phase, that the best-developed machine learning model is the SVM (SVM_n log_) model, which has a value of 0.243 followed by the ANN (ANN_lin_) model (0.306), which is the random forest model and the model with the highest RMSE value for this phase (0.373). It is clear that, for this phase, the three selected models present suitable adjustments. In addition, these models also present high values of the correlation coefficient, all greater than 0.990. These promising results are also obtained for the training phase, although the random forest model presents an important increase regarding RMSE (from 0.373 to 0.824). 

For the query phase, the RMSE values obtained by the model show an increase that, in the same way, happened for the models with the input variables shown in selection Type 1. In addition, looking at the data for the query phase of Table 2 and Table 3, it can be seen that the incorporation of the five variables concerning the input variables’ selection Type 1 destabilizes the models’ prediction, causing in all of them an increase in the RMSE value for this phase. 

Despite this, the random forest and support vector machine models improve the results of the three-variable model proposed by Li et al. (2020) (0.693, 0.443 vs. 0.752, respectively, in terms of RMSE values for the test phase). The artificial neural network model developed with seven input variables presents an RMSE value close to the value of the Li et al. (2020) model for the query phase (0.762 vs. 0.752). Only the SVM model developed using the input variables selection Type 2 has improved the results over the ML models that used the input variables’ selection Type 1.

The second group of models (PE/freshwater) corresponds to machine learning models aimed at predicting the adsorption capacity of polyethylene in freshwater using eight input variables (log *D*, *M*′_w_, *ε*_α_, *ε*_β_, *q*H^+^, *q^−^*, *V*′, *π*). In this case, the best model, based on the RMSE value for the validation phase, corresponds to the ANN (ANN_log_) model (0.446), followed by the SVM (SVM_n_) model (0.473) and the RF model (0.697). These reasonable adjustments are reflected in the high correlation coefficients all greater than 0.960. This behavior is improved in all statistical parameters for the training phase, except for the mean absolute percentage error of the random forest model. In the case of the query phase, these new models present RMSE values between 0.210 and 0.392, maintaining high correlation coefficients, all higher than 0.980. Comparing the ML models developed using the input variables selection Type 2 with the previously developed models using the input variables selection Type 1, it can be said that the ML models developed with eight variables improve the models developed with only one variable; the improvement is appreciable in all the parameters except three MAPE values.

Because of the results reported in Table 3, it can be concluded that the RF, SVM, and ANN models developed using eight input variables improve the model developed by Li et al. (2020) (0.392, 0.210, and 0.272 vs. 0.661, respectively, in terms of RMSE values for test phase). 

The next group of models (PE/pure water) corresponds with ML models to predict the adsorption of polyethylene in pure water. In this case, these models were developed using the eight input variables (log *D*, *M*′_w_, *ε*_α_, *ε*_β_, *q*H^+^, *q^−^*, *V*′, *π)* instead of the two or one which were used by Li et al. (2020) and that was also used in the development of the previous ML models (Table 2). In this case, the optimization process carried out by the RF model involved the elimination of the variable *V*′ in the trees of the forest.

It can be seen in Table 3 that the best-selected model, according to the RMSE value for the validation phase, is the SVM (SVM_log_) model, which presents a value of 0.154, followed by the RF model (0.204) and the ANN (ANN_log_) model (0.403). As in the previous models developed using the input variables selection Type 2, the correlation coefficients are high, all greater than 0.930. This good behavior for the validation phase is also observed in the training phase, although a small increase in the errors made by the models can be seen. For the query phase, the different models present RMSE values between 0.433 and 0.551, keeping the MAPE value at approximately 10% and correlation coefficients greater than 0.920.

Comparing the ML models Type 2 with the previously developed models Type 1, it can be said that, for the query phase, the random forest and support vector machine models present similar adjustments in terms of RMSE to those presented by the Type 1 models. Despite this, only the support vector machine model improves the results of the best model proposed by Li et al. (2020) (0.433 vs. 0.471, respectively, in terms of RMSE values for the test phase). 

The next models (PP/seawater) correspond to the models developed to predict the adsorption of polypropylene in seawater using seven input variables (log *D*, *M*′_w_, *ε*_α_, *ε*_β_, *q^−^*, *V*′, *π*).

Based on the results provided by the root mean square error in the validation phase, it can be said that the best model is the support vector machine (SVM_log_) model (0.229), followed by the random forest model (0.245), and finally, the artificial neural network (ANN_lin_) model, which presents a higher error than the other two models (0.419). The correlation coefficients of the three models are greater than 0.975. This good behavior in the validation phase is also observed in the training phase, for both the random forest model and the support vector machine model; however, it should be noted that the artificial neural network model presents an error of 0.029 in the training phase. The three models present RMSEs for the query phase between 0.215 and 0.494, with the support vector machine model offering the best results, as was the case in the validation phase.

If the results obtained by the models developed using the input variables selection Type 2 are compared with Type 1, it can be said that the increase in the number of variables has led to a significant decrease in the RMSE values obtained in the query phase for the RF and the SVM models. This can be seen in the support vector machine model, which goes from an RMSE of 0.779 to 0.240.

Given the results reported in Table 3, it can be concluded that the RF and the SVM models developed using seven input variables improve the model developed by Li et al. (2020) with two variables (0.215 and 0.240 vs. 0.369, respectively, in terms of RMSE values for test phase). In addition, these models also improve the machine learning models developed using the input variables selection Type 1 except for the ANN model, which is slightly worse.

Finally, the last group of models (PS/seawater) corresponds to the ML models to predict the adsorption for polystyrene in seawater using seven input variables (log *D*, *M*′_w_, *ε*_α_, *ε*_β_, *q^−^*, *V*′, *π).* In these new models, a significant improvement can be seen in the validation and query phase adjustment parameters. In fact, for the validation phase, the RMSE values are between 0.290 and 0.475 for the SVM (SVM_n2 log_) model and the RF model, respectively, while in the Type 1 models, the RMSE values were included between 0.524 and 0.794. Similar behavior is observed for the query phase, with the RMSE values between 0.385 and 0.873. As can be seen in Table 3, the best model on this occasion is the support vector machine model, which also offers the best adjustment parameters for the query phase (0.385).

Given the results, it can be said that the SVM and the ANN (ANN_log_) models developed using seven input variables improve the model developed by Li et al. (2020) with two input variables (0.385 and 0.407 vs. 0.714, respectively, in terms of RMSE values for the test phase).

Figure 2 represents the experimental and predicted values of log *K*_d_ for the best machine learning models according to RMSE in the validation phase of each block shown in Table 3. 

Each graph shows that the adjustments of the training, validation, and query cases are conveniently fitted to the line of slope 1, although some deviation can be observed as it happens in a query case for the PE/seawater model or the PE/pure water model. In general, it can be seen that all the best models consistently predict the log *K*_d_ values.

Given the results shown in Table 1 and Table 2, key points can be drawn about the results obtained for the different machine learning models developed.

Regardless of the input variables chosen, there is always some machine learning model that improves (in terms of RMSE for the query phase) the good adjustments provided by the models developed by Li et al. (2020).Including additional variables to develop the ML models does not always improve the variable selection carried out by Li et al. (2020). This is especially evident in the ML models destined to predict PE/seawater, where no model developed using the input variables selection Type 2 improves the Type 1 models. In this sense, it can be said that the selection of variables carried out by Li et al. (2020) is a good and reliable selection for the model’s development.Both models developed by Li et al. (2020) and the models developed in this research are models that could be used to determine the log *K*_d_ values.To the best of the authors’ knowledge, increasing the number of experimental cases for each microplastic/water group used to develop the models would be appropriate. Presumably, this increase would help the models present better adjustments.

## 4. Conclusions

In this research, various prediction models based on machine learning were developed using different variables to determine the adsorption capacity for PE, PP, and PS towards organic pollutants in various specific water environments.

Given the results, it can be concluded, regardless of the variables chosen for the development of the model, that there is always some machine learning model that provides good results. 

On the other hand, the increase in input variables does not necessarily mean an improvement in the results of the models. This can be seen in the models intended to be used in PE/seawater, where no model developed using the variables selection Type 2 improves the Type 1 models.

To the best of the authors’ knowledge, it would be necessary to improve all models using: i) more experimental cases for each microplastic/water group; and ii) different datasets for training, validation, and query, or the means of different configuration parameters, among others.

## Figures and Tables

**Figure 1 nanomaterials-13-01061-f001:**
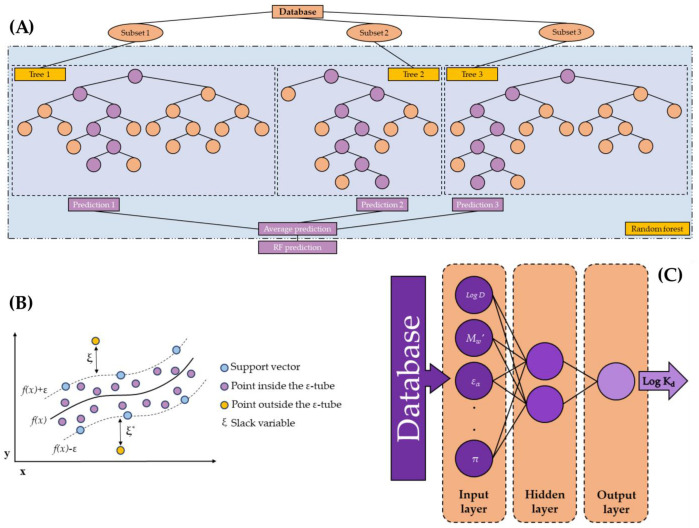
Schemes of the different ML models developed in this research: (**A**) RF model (in regression mode)—inspired by the figure of Yang et al. (2019) [55]; (**B**) SVM model—inspired by the figure of Sarraf Shirazi and Frigaard (2021) [56]; and (**C**) ANN model—inspired by the figures of Moldes et al. (2016) and Zou et al. (2021) [57,58].

**Figure 2 nanomaterials-13-01061-f002:**
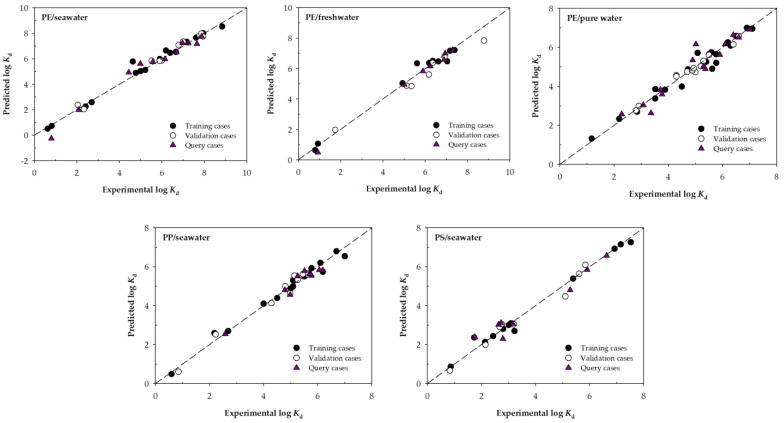
Scatter plots for the experimental and predicted values of log *K*_d_ for the selected ML models developed using the input variables selection Type 2. The dashed line corresponds to the line with slope 1.

**Table 1 nanomaterials-13-01061-t001:** Input variables, marked in purple, are used according to the input variable selection to predict log *K*_d_. Type 1 and type 1 * are the configurations used by Li et al. (2020) [17], and Type 2 is the configuration used in this research. Polyethylene (PE), polystyrene (PS), polypropylene (PP), and the eight variables reported used by Li et al. (2020) [17]: (i) n-octanol/water distribution coefficient at special pH condition (log *D*); (ii) molecular mass (*M*′_w_); covalent; (iii) acidity (*ε*_α_); and (iv) basicity (*ε*_β_); (v) most positive atomic charge on H atom (*q*H^+^); (vi) most negative atomic charge (*q^−^*); (vii) molecular volume (*V*′); and (viii) molecular volume (*π*).

Model	Input Variables								Model	Input Variables							
	log *D*	*M’* _w_	*ε* _α_	*ε* _β_	*q*H^+^	*q^−^*	*V*′	*π*		log *D*	*M’* _w_	*ε* _α_	*ε* _β_	*q*H^+^	*q^−^*	*V*′	*π*
**PE/seawater**									**PP/seawater**								
**Type 1**									**Type 1**								
**Type 2**									**Type 2**								
**PE/freshwater**									**PS/seawater**								
**Type 1**									**Type 1**								
**Type 2**									**Type 2**								
**PE/pure water**																	
**Type 1**																	
**Type 1 ***																	
**Type 2**																	

**Table 2 nanomaterials-13-01061-t002:** Adjustments for the different machine learning models developed using the input variables selection Type 1. RMSE is the root mean square error; MAPE corresponds to the mean absolute percentage error; and r is the correlation coefficient. RF is the random forest model; SVM is the support vector machine model; and ANN corresponds to the artificial neural network model. T, V, and Q are the training, validation, and query phases, respectively. The best models (regarding RMSE for the validation phase) are in bold.

	T			V			Z		
Model	RMSE	MAPE	r	RMSE	MAPE	r	RMSE	MAPE	r
**PE/seawater**									
**RF**	0.525	18.67	0.983	0.380	7.48	0.988	0.523	13.38	0.979
**SVM**	0.287	2.83	0.993	0.248	4.61	0.993	0.357	13.24	0.990
**ANN**	**0.257**	**3.13**	**0.994**	**0.236**	**4.42**	**0.994**	**0.561**	**23.33**	**0.979**
**PE/freshwater**									
**RF**	**0.549**	**8.08**	**0.973**	**0.744**	**13.67**	**0.944**	**0.565**	**7.23**	**0.963**
**SVM**	0.536	8.93	0.976	0.770	11.14	0.945	0.475	10.46	0.978
**ANN**	0.489	6.79	0.978	0.865	13.20	0.932	0.464	8.59	0.974
**PE/pure water—1**									
**RF**	0.471	11.28	0.968	0.176	3.31	0.992	0.531	9.48	0.929
**SVM**	**0.356**	**5.93**	**0.974**	**0.132**	**2.06**	**0.993**	**0.411**	**6.90**	**0.958**
**ANN**	0.309	4.92	0.981	0.225	3.92	0.982	0.729	12.21	0.937
**PE/pure water—2**									
**RF**	**0.410**	**7.79**	**0.967**	**0.132**	**2.25**	**0.993**	**0.526**	**8.59**	**0.936**
**SVM**	0.466	9.51	0.955	0.205	3.47	0.983	0.439	8.10	0.953
**ANN**	0.409	6.45	0.965	0.231	4.23	0.981	0.431	7.72	0.955
**PP/seawater**									
**RF**	**0.255**	**9.95**	**0.990**	**0.199**	**6.69**	**0.994**	**0.298**	**4.97**	**0.968**
**SVM**	0.260	5.12	0.989	0.244	6.92	0.988	0.779	7.32	0.817
**ANN**	0.160	3.19	0.996	0.270	8.94	0.988	0.307	4.21	0.956
**PS/seawater**									
**RF**	0.221	5.28	0.996	0.794	14.61	0.883	1.003	15.11	0.820
**SVM**	**0.554**	**23.10**	**0.969**	**0.524**	**21.69**	**0.965**	**0.436**	**12.85**	**0.988**
**ANN**	0.337	9.21	0.988	0.643	15.69	0.972	0.773	15.07	0.956

**Table 3 nanomaterials-13-01061-t003:** Adjustments for the different machine learning models developed using the input variables selection Type 2. RMSE is the root mean square error, MAPE corresponds to the mean absolute percentage error, and r is the correlation coefficient. RF is the random forest model, SVM is the support vector machine model, and ANN corresponds to the artificial neural network model. T, V, and Q are the training, validation, and query phases, respectively. The best models (regarding RMSE for the validation phase) are in bold.

	T			V			Z		
Model	RMSE	MAPE	r	RMSE	MAPE	r	RMSE	MAPE	r
**PE/seawater**									
**RF**	0.824	38.89	0.954	0.373	7.69	0.988	0.693	26.80	0.970
**SVM**	**0.336**	**5.52**	**0.991**	**0.243**	**5.22**	**0.994**	**0.443**	**16.38**	**0.984**
**ANN**	0.040	0.56	1.000	0.306	5.46	0.989	0.762	15.28	0.946
**PE/freshwater**									
**RF**	0.424	16.78	0.991	0.697	8.78	0.962	0.392	11.86	0.986
**SVM**	0.320	6.87	0.991	0.473	7.05	0.990	0.210	8.18	0.999
**ANN**	**0.289**	**4.94**	**0.992**	**0.446**	**7.10**	**0.991**	**0.272**	**10.40**	**0.997**
**PE/pure water**									
**RF**	0.473	10.77	0.955	0.204	3.31	0.983	0.542	10.37	0.929
**SVM**	**0.306**	**5.34**	**0.981**	**0.154**	**2.56**	**0.990**	**0.433**	**7.25**	**0.956**
**ANN**	0.634	14.70	0.916	0.403	7.90	0.937	0.551	11.57	0.926
**PP/seawater**									
**RF**	0.295	6.44	0.988	0.245	9.42	0.994	0.215	3.36	0.983
**SVM**	**0.222**	**4.74**	**0.992**	**0.229**	**6.98**	**0.990**	**0.240**	**3.66**	**0.974**
**ANN**	0.029	0.54	1.000	0.419	12.20	0.979	0.494	8.20	0.938
**PS/seawater**									
**RF**	0.486	11.07	0.980	0.475	15.16	0.970	0.873	23.01	0.882
**SVM**	**0.248**	**4.72**	**0.994**	**0.290**	**8.50**	**0.986**	**0.385**	**12.05**	**0.976**
**ANN**	0.309	7.01	0.990	0.445	9.74	0.984	0.407	12.43	0.973

## Data Availability

The data used were taken from the research paper of Li et al. (2020).

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
