# Peer review of "Machine Learning to Predict the Adsorption Capacity of Microplastics"

_nanomaterials, 2023, doi:10.3390/nano13061061_

Round 1

Reviewer 1 Report

The work further develops the methodologies presented by Hsu et al. (2016) and Li et al. (2020). The work specifies and refines the methodologies presented by Hsu et al. (2016) and Li et al. (2020). Improved by the authors three machine learning models could be used for rapid estimation of the absorption of organic contaminants on microplastics. The authors propose to improve all models using: a) more experimental cases for each microplastic/water group, b) different datasets for training, validation, and query, or means different configuration parameters, among others.

Dear authors, my recommendation is that the paper should be accepted in the journal Nanomatetials. I have only one small suggestion for you:

 Page 14, Lines 504-505, In the sentence: Figure 1 represents the experimental and predicted values of log Kd for the best machine learning models, according to RMSE in the validation phase) of each block shown in Table 3.

 Comment: This should be Figure 2.

Author Response

Dear reviewer,

Thank you very much for your kind words; we have made the changes you requested in your review.

Comments and Suggestions for Authors

The work further develops the methodologies presented by Hsu et al. (2016) and Li et al. (2020). The work specifies and refines the methodologies presented by Hsu et al. (2016) and Li et al. (2020). Improved by the authors three machine learning models could be used for rapid estimation of the absorption of organic contaminants on microplastics. The authors propose to improve all models using: a) more experimental cases for each microplastic/water group, b) different datasets for training, validation, and query, or means different configuration parameters, among others.

Dear authors, my recommendation is that the paper should be accepted in the journal Nanomatetials. I have only one small suggestion for you:

Page 14, Lines 504-505, In the sentence: Figure 1 represents the experimental and predicted values of log Kd for the best machine learning models, according to RMSE in the validation phase) of each block shown in Table 3.

Comment: This should be Figure 2.

Answer: The reviewer is right; we appreciate your feedback and have made the suggested changes.

Reviewer 2 Report

Authors have presented an article entitled “Machine learning to predict the adsorption capacity for micro-plastics” Though the manuscript is well written and organized but there is scope for further improving the quality of the draft before considering for publication.  

Few minor comments are listed below:

1. As author mentioned that the fibers and textiles are used in many areas such as packaging, electronic industries etc. as micro domain. Author should send the performance of other articles to make the contrast: Composites Part A: Applied Science and Manufacturing, 107427 (2023); Fibers and Polymers 19 (5), 1064-1073 (2018). 

2. In page 13, line no. 505, there is one bracket after the validation phase); please check and rectify.

3. In Figure 1, the author should use figure numbers such as (a), (b)… and use the figure caption to highlight it. It will be helpful to the readers.   

4. Authors nicely designed the results and discussion section, they should check the typos and grammatical mistakes in the revised version.  

Author Response

Dear reviewer,

Thank you very much for your kind words; we have made the changes you requested in your review.

Comments and Suggestions for Authors

Authors have presented an article entitled “Machine learning to predict the adsorption capacity for micro-plastics” Though the manuscript is well written and organized but there is scope for further improving the quality of the draft before considering for publication. 

Few minor comments are listed below:

  1. As author mentioned that the fibers and textiles are used in many areas such as packaging, electronic industries etc. as micro domain. Author should send the performance of other articles to make the contrast: Composites Part A: Applied Science and Manufacturing, 107427 (2023); Fibers and Polymers 19 (5), 1064-1073 (2018).

Answer: The suggested articles have been conveniently included in the revised manuscript.

Page 2, lines 31-32: The use of microbeads or nanobeads based on plastic polymers (e.g., Bisphenol-A diglycidyl ether, polyetheramine, and Polyvinyl alcohol) to functionalize and decorate CNTs has been reported to improve their shielding against electromagnetic interference, as well as, the electrical and mechanical properties of elastic, functional composites can be improved by interlacing the beaded and coated fibers into a smart tissue [2,3].

  1. In page 13, line no. 505, there is one bracket after the validation phase); please check and rectify.

Answer: The reviewer is correct; the bracket has been conveniently removed.

  1. In Figure 1, the author should use figure numbers such as (a), (b)… and use the figure caption to highlight it. It will be helpful to the readers.

Answer: As suggested by the reviewer, the different parts of the figure have been highlighted, by placing (A), (B) and (C), both in the figure and its caption.

  1. Authors nicely designed the results and discussion section, they should check the typos and grammatical mistakes in the revised version.

Answer: Grammatical and typographical errors have been revised accordingly at the reviewer's request.